# Key Taxa of the Gut Microbiome Associated with the Relationship Between Environmental Sensitivity and Inflammation-Related Biomarkers

**DOI:** 10.3390/microorganisms13010185

**Published:** 2025-01-16

**Authors:** Satoshi Takasugi, Shuhei Iimura, Miyabi Yasuda, Yoshie Saito, Masashi Morifuji

**Affiliations:** 1R&D Division, Meiji Co., Ltd., 1-29-1 Nanakuni, Hachioji 192-0919, Japan; 2Faculty of Education, Soka University; 1-236 Tangi-machi, Hachioji 192-8577, Japan; iimurashuhei@gmail.com; 3Wellness Science Labs, Meiji Holdings Co., Ltd., 1-29-1 Nanakuni, Hachioji 192-0919, Japan; miyabi.yasuda@meiji.com (M.Y.); yoshie.saitou@meiji.com (Y.S.); masashi.morifuji@meiji.com (M.M.)

**Keywords:** environmental sensitivity, sensory processing sensitivity, gut microbiome, inflammation, gut barrier, gut-brain axis, *Butyricimonas*, *Coprobacter*, *Akkermansia*

## Abstract

Individual differences in environmental sensitivity are linked to stress-related psychiatric symptoms. In previous research, we found that high environmental sensitivity can be a risk factor for increased inflammation and gut permeability, particularly when gut microbiome diversity is low. However, the specific gut bacterial taxa involved in this interaction remain unclear. As a preliminary study, this research aimed to identify the key gut microbiome taxa associated with this relationship. Environmental sensitivity, gut microbiome composition, gut permeability (lipopolysaccharide-binding protein, LBP), and inflammation (C-reactive protein, CRP) biomarkers were evaluated in 88 participants. The interaction between environmental sensitivity and the relative abundance of the family *Marinifilaceae* (genus *Butyricimonas*) was a predictor of CRP levels. Similarly, the interaction between environmental sensitivity and relative abundance of the family *Barnesiellaceae* (genus *Coprobacter*), the family *Akkermansiaceae* (genus *Akkermansia*), the genus *Family XIII AD3011 group*, the genus *GCA-900066225*, or the genus *Ruminiclostridium 1* predicted LBP levels. Individuals with high environmental sensitivity exhibited elevated CRP or LBP levels when the relative abundance of these taxa was low. Conversely, highly sensitive individuals had lower CRP or LBP levels when the relative abundance of these taxa was high. This study suggests that specific taxa serve as one of the protective factors against inflammation and gut permeability in individuals with high environmental sensitivity. Further in-depth studies are needed to confirm these associations and understand the underlying mechanisms.

## 1. Introduction

Environmental sensitivity, also known as sensory processing sensitivity (SPS), is a genetic, biological, and psychological trait that accounts for individual differences in how people perceive and process both negative and positive environments [1]. Some individuals are more sensitive to environmental influences due to their genetic or temperamental predispositions. As a result, they are more negatively affected by stressful environments and more positively impacted by supportive ones. This suggests that individuals with high environmental sensitivity are more likely to be influenced “for better and for worse” by both positive and negative environments [2,3,4].

Examining the darker aspects of this concept, it has been reported that high environmental sensitivity is linked to both physical and psychiatric symptoms. In a previous study involving a large sample size, we demonstrated that individuals with higher environmental sensitivity were more likely to self-report gastrointestinal symptoms, even after adjusting for sociodemographic characteristics [5]. Additionally, several studies have found positive associations between environmental sensitivity and physical symptoms [6,7], such as back pain, diarrhea, heartburn, and sore throat [8], as well as cardiovascular, respiratory, and gastrointestinal issues [9]. Conversely, one study indicated that there was no clear correlation between SPS and somatic symptoms [10]. Consequently, it remains a topic of debate whether environmental sensitivity is associated with physical health.

Accumulating evidence indicates that individuals with high environmental sensitivity are not only more vulnerable to worsening psychopathology in stressful environments than those with low sensitivity, but they are also less likely to experience worsening psychopathology if they can enhance protective factors (i.e., vantage sensitivity) [11,12,13]. Thus, the relationship between environmental sensitivity and physical/mental health is influenced by the quality of internal or external environmental factors.

In a previous study, we explored the correlations among individual differences in environmental sensitivity, inflammation-related biomarkers (C-reactive protein, CRP, and lipopolysaccharide-binding protein, LBP), and gut microbiome diversity as one of the internal environmental factors. CRP is recognized as a risk factor associated with various psychological and physical symptoms, including cognitive function [14,15], depression [16], cardiovascular disease [17,18,19,20,21,22], and irritable bowel syndrome (IBS) [23]. Our study revealed that the relationship between environmental sensitivity and these inflammation-related biomarkers (CRP and LBP) was not merely a simple cause-and-effect relationship; rather, the levels of both biomarkers can vary, depending on the interaction between environmental sensitivity and gut microbiome diversity [24]. Specifically, individuals with higher environmental sensitivity and lower gut microbiome diversity exhibited elevated levels of both biomarkers. In contrast, higher gut microbiome diversity did not elevate the levels of either biomarker, even among highly susceptible individuals. These findings suggest that individual differences in environmental sensitivity may play a role in the brain–gut–microbiome interaction, with gut microbiome diversity serving a protective function against inflammatory responses in individuals with high environmental sensitivity.

However, the role of specific gut bacterial taxa in moderating the link between environmental sensitivity and inflammatory reactions is not yet fully understood. Addressing this gap will enhance our understanding of why individuals with high environmental sensitivity and low gut microbiota diversity exhibit heightened inflammatory responses. Therefore, as a preliminary study, this research aims to identify for the first time key taxa in the gut microbiome that are associated with the relationship between environmental sensitivity and inflammation biomarkers.

## 2. Materials and Methods

### 2.1. Study Procedure

This study was a follow-up analysis of a previous report [24]. Initially, we recruited 110 adults who had previously participated in another study measuring fecal and/or blood biomarkers. Informed consent to participate in this study and a complete additional questionnaire was obtained from 90 of these individuals. The participants underwent a physical examination, blood tests, and fecal microbiome analysis. Two years later, they completed a web-based questionnaire. Fecal samples were unavailable from two participants, so data from 88 participants were included in the analysis. This study was preregistered with the University Hospital Medical Information Network (https://www.umin.ac.jp/english/, accessed on 10 October 2024) (Identifier: UMIN000047571) and was approved by the Ethics Committee of Meiji Co., Ltd. (Tokyo, Japan) Institutional Review Board (No. 2021-012) on 24 February 2022, in accordance with the guidelines of the Declaration of Helsinki. All participants signed the informed consent form digitally.

### 2.2. Environmental Sensitivity

To measure the personality traits associated with environmental sensitivity, we utilized the 10-item Japanese version of the Highly Sensitive Person Scale (HSP-J10) [25]. This scale evaluates susceptibility to both negative and positive environmental influences and includes the following items: “Do changes in your life shake you up?”, “Are you easily overwhelmed by strong sensory input?”, “Do other people’s moods affect you?”, “Do you get rattled when you have a lot to do in a short amount of time?”, “When you must compete or be observed while performing a task, do you become so nervous or shaky that you do much worse than you would otherwise?”, “Are you bothered by intense stimuli, like loud noises or chaotic scenes?”, “Are you made uncomfortable by loud noises?”, “Are you easily overwhelmed by things like bright lights, strong smells, coarse fabrics, or sirens close by?”, “Do you notice and enjoy delicate or fine scents, tastes, sounds, works of art?”, and “Are you deeply moved by the arts or music?”. Each item was rated on a 7-point Likert-type scale, ranging from 1 (strongly disagree) to 7 (strongly agree). Cronbach’s alpha, a measure of the scale’s internal consistency, was adequate at 0.85 (McDonald’s omega total = 0.86). The mean scores of the 10 items were used for the analyses. Personality traits, including environmental sensitivity [26], are psychological factors that tend to remain stable within individuals over time [27].

### 2.3. Blood Biomarkers

Fasting venous blood was collected, and serum was obtained through centrifugation and stored at −80 °C until analysis. Serum high-sensitivity CRP, an inflammation biomarker, was measured using the V-PLEX Vascular Injury Panel 2 Human Kit (Meso Scale Diagnostics, Rockville, MD, USA). Serum LBP, a gut permeability biomarker [28,29], was measured using an LBP Human ELISA Kit (Hycult Biotech, Uden, The Netherlands). Both biomarker data were log-transformed prior to analysis, as previously reported [30,31], due to the data not being normally distributed.

### 2.4. Gut Microbiome

Fecal samples were collected and prepared as previously described [24]. Briefly, participants collected feces on any day during this study. The fecal samples were homogenized using a FastPrep-24 5G (MP Biomedicals, Irvine, CA, USA) with 0.1 mm zirconia beads (EZ-Extract for DNA/RNA, AMR, Tokyo, Japan). DNA was then extracted from the fecal samples using the QIAamp DNA Stool Mini Kit (QIAGEN, Hilden, Germany) following “Protocol Q” [32], with minor modifications. The V3–V4 region of the 16S ribosomal RNA gene was amplified by PCR using universal bacterial primer sets (5′-TCGTCGGCAGCGTCAGATGTGTATAAGAGACAGCCTACGGGNGGCWGCAG-3′ and 5′-GTCTCGTGGGCTCGGAGATGTGTATAAGAGACAGGACTACHVGGGTATCTAATCC-3′) and sequenced using the MiSeq Reagent kit v3 (600 cycle) (Illumina Inc., San Diego, CA, USA). The sequence data were processed using QIIME2 (version 2020.2), which included quality filtering and the identification of amplicon sequence variants (ASVs) using the DADA2 algorithm. For downstream analysis, 32,000 reads were randomly selected from each sample. Taxonomic classification was performed by conducting BLAST searches of representative ASV sequences against the SILVA 132 database. The ratios of each family and genus, as well as the alpha diversity indices, were calculated using QIIME2. Alpha diversity was evaluated for richness using observed operational taxonomic units (OTUs; observed species) and for biodiversity using Faith’s phylogenetic diversity (PD) [33]. The relative abundance of the gut microbiome at the family and genus levels was calculated. Taxa with very low prevalence at the family or genus level (<5%) were excluded from further analyses.

### 2.5. Data Analyses

The statistical significance level for the series of analyses was set at α = 0.05. All statistical analyses were conducted using R version 4.4.1 [34] and its interface, RStudio version 2024.04.2 [35]. Prior to the main analysis, gender differences in the taxa of the gut microbiome were also tested.

In the previous study, we found that the interactions between environmental sensitivity and alpha diversity indices (OTUs and PD) of the gut microbiome accounted for the levels of CRP and LBP. Therefore, we initially conducted a Pearson correlation analysis to examine the relationship between alpha diversity indices and gut microbiome taxa at the family and genus levels to investigate the key taxa of the gut microbiome associated with the interaction between environmental sensitivity and these biomarkers.

Second, a series of hierarchical multiple regression analyses were conducted to identify the interaction effect between environmental sensitivity and specific gut microbiome taxa. The interaction effect analysis was conducted solely for the gut microbiome taxa at the family level, where significant moderate (|*r*| > 0.30) or high (|*r*| > 0.50) correlations were observed. Following this, interaction effect analysis was performed at the genus level for both those bacterial taxa that exhibited significant interactions at the family level or significant moderate (|*r*| > 0.30) or high (|*r*| > 0.50) correlations at the genus level. The interaction effects of environmental sensitivity and gut microbiome taxa, at both the family and genus levels, on CRP and LBP were examined using hierarchical multiple regression analysis [36]. As previously reported [24], in Step 1, sex, age, BMI, environmental sensitivity, and the relative abundance of gut microbiome taxa at the family or genus level were entered as independent variables, while the two biomarkers (CRP and LBP) were entered as dependent variables. In Step 2, an interaction term representing the relationship between environmental sensitivity and the relative abundance of gut microbiome taxa was added to the independent variables entered in Step 1.

Finally, if the interaction term was significant in Step 2, a simple slope test was conducted to assess the moderating effect of the relative abundance of the gut microbiome on the relationship between environmental sensitivity and the biomarkers. Regression coefficients were estimated by substituting M − 1SD, mean, and M + 1SD for the gut microbiome taxa at either the family or genus level in the model, respectively [36].

## 3. Results

### 3.1. Clinical and Biochemical Characteristics of the Subjects

The clinical and biochemical characteristics of the 88 subjects from whom fecal samples were collected are presented in Table 1.

### 3.2. Relative Abundance and Prevalence of Gut Microbiome Taxa, Along with the Correlations Between Alpha Diversity Indices and Bacterial Taxa

The relative abundance and prevalence of gut microbiome taxa at the family and genus levels are presented in Table 2 and Table 3, respectively, both of which exclude taxa with a prevalence of less than 5%.

Females showed significantly higher means of relative abundance in *Eggerthellaceae*, *Coriobacteriales unclassified family*, *Marinifilaceae*, *Porphyromonadaceae*, *Rikenellaceae*, *Christensenellaceae*, *Clostridiales vadin BB60 group*, *Defluviitaleaceae*, *Eubacteriaceae*, *Clostridiales Family XIII*, and *Ruminococcaceae* and *Akkermansiaceae* at the family level, as well as in *Varibaculum*, *Gordonibacter*, *Coriobacteriales unclassified family unclassified genus*, *Butyricimonas*, *Porphyromonas*, *Alistipes*, *Christensenellaceae R-7 group*, *Christensenellaceae uncultured bacterium*, *Christensenellaceae unclassified genus*, *Clostridiales vadin BB60 group uncultured bacterium*, *Defluviitaleaceae UCG-011*, *Anaerofustis*, *Eubacterium*, *Anaerococcus*, *Family XIII AD3011 group*, *[Eubacterium] brachy group*, *Eisenbergiella*, *GCA900066575*, *Lachnospiraceae NK4A136 group*, *Shuttleworthia*, *Peptococcaceae uncultured bacterium*, *Anaerofilum*, *Anaerotruncus*, *DTU089*, *Oscillibacter*, *Ruminiclostridium*, *Ruminococcaceae UCG-005*, *Subdoligranulum*, *UBA1819*, *Ruminococcaceae uncultured bacterium*, and *Akkermansia* at the genus level than males. Males showed significantly higher means of relative abundance in *Veillonellaceae* at the family level, as well as in *Actinomyces*, *Dorea*, *Lachnoclostridium*, *Lachnospiraceae UCG-004*, *Lachnospiraceae UCG-010*, *Butyricicoccus*, and *Megasphaera* at the genus level than females.

We identified significant moderate to high correlations between alpha diversity indices (OTUs and/or PD) and gut microbiome taxa at the family level in the following 17 taxa: *Methanobacteriaceae*, *Coriobacteriales Incertae Sedis*, *Coriobacteriales unclassified family*, *Barnesiellaceae*, *Marinifilaceae*, *Rikenellaceae*, *Christensenellaceae*, *Clostridiales vadinBB60 group*, *Defluviitaleaceae*, *Family XIII (Clostridiales)*, *Lachnospiraceae*, *Ruminococcaceae*, *Clostridiales unclassified family*, *DTU014 uncultured bacterium*, *Synergistaceae*, *Mollicutes RF39 uncultured bacterium*, and *Akkermansiaceae*. Furthermore, we identified significant moderate to high correlations between alpha diversity indices (OTUs and/or PD) and gut microbiome taxa at the genus level in the following 63 taxa: *Methanobrevibacter*, *Varibaculum*, *Enterorhabdus*, *Senegalimassilia*, *Coriobacteriales unclassified family unclassified genus*, *Barnesiella*, *Butyricimonas*, *Odoribacter*, *Alistipes*, *Christensenellaceae R-7 group*, *Christensenellaceae uncultured bacterium*, *Christensenellaceae unclassified genus*, *Clostridiales vadinBB60 group uncultured bacterium*, *Defluviitaleaceae UCG-011*, *Family XIII AD3011 group*, *Family XIII UCG-001*, *Coprococcus 1*, *Coprococcus 2*, *Eisenbergiella*, *GCA900066575*, *Lachnoclostridium*, *Lachnospiraceae FCS020 group*, *Lachnospiraceae NK4A136 group*, *Marvinbryantia*, *Sellimonas*, *Shuttleworthia*, *[Ruminococcus] gnavus group*, *Peptococcaceae uncultured bacterium*, *Romboutsia*, *Anaerofilum*, *Anaerotruncus*, *Butyricicoccus*, *DTU089*, *GCA-900066225*, *Hydrogenoanaerobacterium*, *Negativibacillus*, *Papillibacter*, *Ruminiclostridium*, *Ruminiclostridium 1*, *Ruminococcaceae NK4A214 group*, *Ruminococcaceae UCG-002*, *Ruminococcaceae UCG-005*, *Ruminococcaceae UCG-007*, *Ruminococcaceae UCG-009*, *Ruminococcaceae UCG-010*, *Ruminococcaceae UCG-014*, *Ruminococcus 1*, *Ruminococcus 2*, *UBA1819*, *[Eubacterium] coprostanoligenes group*, *Ruminococcaceae unclassified genus*, *Clostridiales unclassified family unclassified genus*, *DTU014 uncultured bacterium uncultured bacterium*, *Holdemania*, *Erysipelotrichaceae unclassified genus*, *Veillonella*, *Coriobacteriales Incertae Sedis uncultured bacterium*, *Desulfovibrio*, *Desulfovibrionaceae uncultured bacterium*, *Burkholderiaceae unclassified genus*, *Cloacibacillus*, *Mollicutes RF39 uncultured bacterium uncultured bacterium*, and *Akkermansia.*

### 3.3. Interaction Between Environmental Sensitivity and Gut Microbiome Taxa at the Family Level

Effect on CRP

We examined the interaction effects using hierarchical multiple regression analysis exclusively for the gut microbiome taxa that exhibited significant moderate (|*r*| > 0.30) or high (|*r*| > 0.50) correlations. Among the 17 taxa, the interaction term between environmental sensitivity and the family *Marinifilaceae* was identified as a significant predictor for CRP (*β* = −0.183, *p* < 0.05) (Table 4). The coefficient of determination, *R*^2^, increased from that observed in Step 1 when the interaction term between environmental sensitivity and the family *Marinifilaceae* was incorporated in Step 2 (Δ*R*^2^ = 0.033, *p* < 0.05). The coefficient of determination for the final regression model in Step 2 was *R*^2^ = 0.375 (*p* < 0.01). The interaction between environmental sensitivity and the other bacterial taxa did not show a significant association with CRP.

Effect on LBP

Among the 17 taxa, the interaction term between environmental sensitivity and the family *Barnesiellaceae* emerged as a significant predictor of LBP (*β* = −0.200, *p* < 0.05) (Table 5). The coefficient of determination, *R*^2^, increased from that observed in Step 1 when the interaction term between environmental sensitivity and family *Barnesiellaceae* was included in Step 2 (Δ*R*^2^ = 0.036, *p* < 0.05). The final regression model in Step 2 yielded a coefficient of determination of *R*^2^ = 0.340 (*p* < 0.01). Similar results were observed for the family *Akkermansiaceae* (Δ*R*^2^ = 0.035, *p* < 0.05) (Table 6), family *Marinifilaceae* (Δ*R*^2^ = 0.030, *p* = 0.062) (Table 7), family *Defluviitaleaceae* (Δ*R*^2^ = 0.027, *p* < 0.05) (Table 8), and family *XIII* (Δ*R*^2^ = 0.023, *p* = 0.099) (Table 9). The *p*-values for the interaction between environmental sensitivity and the other bacterial taxa were >0.1 in relation to LBP.

### 3.4. Interaction Between Environmental Sensitivity and Gut Microbiome Taxa at the Genus Level

Effect on CRP

We examined the interaction effects using hierarchical multiple regression analysis for the gut microbiome taxa (genus) within the family where a significant interaction effect was observed (family *Marinifilaceae*) or for the taxa that exhibited significant moderate (|*r*| > 0.30) or high (|*r*| > 0.50) correlations at the genus level. The interaction term between environmental sensitivity and genus *Butyricimonas* emerged as a significant predictor for CRP (*β* = −0.218, *p* < 0.05) (Table 10). The coefficient of determination, *R*^2^, increased from that in Step 1 when the interaction term between environmental sensitivity and genus *Butyricimonas* was added in Step 2 (Δ*R*^2^ = 0.044, *p* < 0.05). The final regression model in Step 2 yielded a coefficient of determination of *R*^2^ = 0.385 (*p* < 0.01). In contrast, the interaction between environmental sensitivity and the other bacterial taxa did not show a significant association with CRP.

Effect on LBP

We conducted hierarchical multiple regression analysis on the gut microbiome taxa (genus) associated with families where a significant interaction effect was observed (family *Barnesiellaceae* and family *Akkermansiaceae*) or for the taxa that exhibited significant moderate (|*r*| > 0.30) or high (|*r*| > 0.50) correlations at the genus level. The interaction between environmental sensitivity and genus *Coprobacter* was found to be a significant predictor for LBP (*β* = −0.233, *p* < 0.05) (Table 11). The coefficient of determination, *R*^2^, increased from that in Step 1 when the interaction term between environmental sensitivity and genus *Coprobacter* was added in Step 2 (Δ*R*^2^ = 0.042, *p* < 0.05). The coefficient of determination for the final regression model, Step 2, was *R*^2^ = 0.331 (*p* < 0.01). Similar findings were observed for genus *Barnesiella* (Δ*R*^2^ = 0.027, *p* = 0.077) (Table 12), genus *Akkermansia* (Δ*R*^2^ = 0.035, *p* < 0.05) (Table 13), genus *Family XIII AD3011 group* (Δ*R*^2^ = 0.053, *p* < 0.05) (Table 14), genus *GCA-900066225* (Δ*R*^2^ = 0.043, *p* < 0.05) (Table 15), and genus *Ruminiclostridium 1* (Δ*R*^2^ = 0.035, *p* < 0.05) (Table 16).

### 3.5. Simple Slope Analysis

We conducted simple slope tests on the gut microbiome taxa that showed an interaction effect with CRP (Figure 1) and LBP (Figure 2). These taxa include family *Marinifilaceae* and genus *Butyricimonas* (associated with CRP), and family *Barnesiellaceae*, family *Akkermansiaceae*, genus *Coprobacter*, genus *Akkermansia*, genus *Family XIII AD3011 group*, genus *GCA-900066225*, and genus *Ruminiclostridium 1* (associated with LBP). 

For individuals with high environmental sensitivity, there was no significant association with CRP when family *Marinifilaceae* abundance was high (*M + 1SD*; *β* = −0.110, *p* = 0.496). However, CRP levels were significantly elevated when the abundance of family *Marinifilaceae* was low (*M − 1SD*; *β* = 0.434, *p* = 0.008) (Figure 1A). Similar results were observed for other taxa. When genus *Butyricimonas* abundance was high (*M + 1SD*; *β* = −0.020, *p* = 0.855), no significant association with CRP was found, but when its abundance was low (*M* − 1*SD*; *β* = 0.392, *p* = 0.003), CRP levels were significantly higher (Figure 1B). 

For LBP, individuals with high environmental sensitivity showed no association when family *Barnesiellaceae* was abundant (*M + 1SD*; *β* = −0.127, *p* = 0.419), but LBP levels were significantly elevated when the abundance of family *Barnesiellaceae* was low (*M − 1SD*; *β* = 0.371, *p* = 0.011) (Figure 2A). Similarly, in individuals with high environmental sensitivity, no association with LBP was found when family *Akkermansiaceae* abundance was high (*M + 1SD*; *β* = −0.143, *p* = 0.386), but LBP levels were elevated when its abundance was low (*M − 1SD*; *β* = 0.362, *p* = 0.015) (Figure 2B). Individuals with high environmental sensitivity showed no association with LBP when genus *Coprobacter* abundance was high (*M + 1SD*; *β* = −0.077, *p* = 0.560), but LBP levels were significantly higher when its abundance was low (*M − 1SD*; *β* = 0.402, *p* = 0.009) (Figure 2C). Similarly, individuals with high environmental sensitivity showed no association with LBP when genus *Akkermansia* abundance was high (*M + 1SD*; *β* = −0.143, *p* = 0.386), but LBP was elevated when genus *Akkermansia* abundance was low (*M − 1SD*; *β* = 0.362, *p* = 0.015) (Figure 2D). Individuals with high environmental sensitivity showed no association with LBP when genus *Family XIII AD3011 group* abundance was high (*M + 1SD*; *β* = −0.126, *p* = 0.376), but LBP was elevated when genus *Family XIII AD3011 group* abundance was low (*M − 1SD*; *β* = 0.393, *p* = 0.004) (Figure 2E). Similarly, individuals with high environmental sensitivity showed no association with LBP when genus *GCA-900066225* abundance was high (*M + 1SD*; *β* = −0.238, *p* = 0.218), but LBP was elevated when genus *GCA-900066225* abundance was low (*M − 1SD*; *β* = 0.420, *p* = 0.006) (Figure 2F). Individuals with high environmental sensitivity showed no association with LBP when genus *Ruminiclostridium 1* abundance was high (*M + 1SD*; *β* = −0.140, *p* = 0.396), but LBP was elevated when genus *Ruminiclostridium 1* abundance was low (*M − 1SD*; *β* = 0.370, *p* = 0.015) (Figure 2G).

## 4. Discussion

This study aimed to investigate the key taxa of the gut microbiome associated with the interaction between environmental sensitivity and inflammation biomarkers. We identified an interaction between environmental sensitivity and the family *Marinifilaceae* (genus *Butyricimonas*), which was associated with CRP, an inflammation biomarker. Additionally, we found interactions between environmental sensitivity and the family *Barnesiellaceae* (genus *Coprobacter*), the family *Akkermansiaceae* (genus *Akkermansia*), the genus *Family XIII AD3011 group*, the genus *GCA-900066225*, and the genus *Ruminiclostridium 1*, all of which are associated with LBP (a gut permeability biomarker). Simple slope tests on these taxa indicated that individuals with high environmental sensitivity did not exhibit significant associations with CRP or LBP when the abundance of these taxa was high. Conversely, elevated levels of these biomarkers were observed when the abundance of the taxa was low. These findings suggest that these taxa play a protective role in moderating inflammation and gut permeability, acting as key components of the gut microbiome that are associated with the interaction between environmental sensitivity and biomarkers of inflammation or gut permeability. Given existing reports linking higher levels of CRP or LBP to increased stress-related psychiatric symptoms, such as depression and anxiety [37,38,39], as well as physical symptoms, including IBS [23], these taxa may play a crucial role in the development or worsening of these symptoms among individuals with high environmental sensitivity.

This study indicated that individual differences in environmental sensitivity may result in different levels of inflammation and gut permeability, depending on the gut microbiome. The serotonin transporter gene polymorphism (5-HTTLPR), one of the genes related to environmental sensitivity, is associated with cortisol responsiveness to acute psychosocial stress [40,41], which suggests that hypothalamic–pituitary–adrenal (HPA) axis responsiveness to the internal and/or external environmental stimuli is involved in individual differences in environmental sensitivity. Several animal and human studies indicate that overactivation of HPA axis in response to chronic stressful stimuli increases gut permeability [42,43,44] and that the gut microbiome can affect the HPA axis [45,46,47,48,49,50,51,52,53], which suggests that the gut microbiome can affect gut permeability through the HPA axis. In contrast, the gut microbiome can also affect gut permeability independently of the HPA axis [54]. With high gut permeability, a lipopolysaccharide (endotoxin) derived from bacteria enters the bloodstream through the intestinal tract and induces inflammation [55]. Therefore, the plausible mechanisms of the interaction between environmental sensitivity and the gut microbiome are as follows. First, the gut microbiome may influence gut permeability and inflammation through the HPA axis as an internal environmental factor; second, the gut microbiome may act as a moderating factor and influence the gut permeability and inflammation caused by HPA axis activation in response to other stressful stimuli.

The genus *Butyricimonas* is a well-known butyrate-producing bacterium in the intestinal tract [56]. Butyrate has been shown to have beneficial effects on both psychological and physical health. In animal models of depression induced by chronic mild stress or maternal deprivation, butyrate exhibited antidepressant effects [57,58]. In vitro and in vivo studies suggest that butyrate may play a significant role in regulating neuromediator gene expression within the enteric nervous system and in gastrointestinal functions, such as motility [59]. Furthermore, butyrate possesses anti-inflammatory properties [60], which operate through several mechanisms [42,55,61,62,63,64,65,66], including the maintenance of the intestinal barrier [55,66] and the attenuation of the HPA axis [42]. Indeed, the present study demonstrated a tendency for interaction between environmental sensitivity and the genus *Butyricimonas*, which is associated with the gut permeability biomarker LBP (Appendix A). Therefore, the various functions of butyrate may help explain how the family *Marinifilaceae* (genus *Butyricimonas*) plays a protective role in inflammation among individuals with high environmental sensitivity.

The genus *Akkermansia* is recognized for its ability to enhance intestinal barrier integrity by stimulating mucin production in both human and animal models [54,67]. Okuma et al. [68] reported that several taxa, including the genera *Coprobacter* and *Butyricimonas*, were found to be less abundant in the male depression group compared to the male control group. The genus *Coprobacter* produces propionic and acetic acids [69]. Both propionic [70] and acetic acids [71] also exhibit protective effects on the intestinal barrier. There are limited reports on the genus *Family XIII AD3011 group*, genus *GCA-900066225*, and genus *Ruminiclostridium 1*. However, Shang et al. [72] reported that the gut microflora, including some taxa, such as *Family XIII AD3011 group*, was conducive to improving disease resistance in pigs. Mao et al. [73] reported that the relative abundance of *GCA-900066225* was positively associated with the cecal short-chain fatty acid levels and colonic gut integrity-related marker in mice. Cao et al. [74] reported that *Ruminiclostridium 1* was negatively correlated with inflammation markers and positively correlated with acetic acid and propionic acid in the ileum of mice. Consequently, these taxa might contribute to gut integrity through these mechanisms in individuals with high environmental sensitivity.

## 5. Possible Limitations and Future Directions

This study has several possible limitations. First, environmental sensitivity was assessed using a questionnaire. Although HSP-J10 is a well-validated scale using a large sample size and experimental manipulation [25], the integration of alternative measurement methods, such as polygenic scores derived from genome-wide analysis [75] or evaluations of neurophysiological reactivity during stressful tasks [76], could provide comprehensive insights and strengthen our findings. Second, this study utilized a cross-sectional design. Generally, cross-sectional study designs are often used to evaluate the associations between variables in microbiome research [77,78,79], as it is technically difficult to reproduce a specific gut microbiome through interventions in humans. Future longitudinal studies would allow for a better exploration of the causal relationships. Third, several factors, such as diet, might affect the results as confounding factors. In particular, higher intakes of fruit and vegetables were reported to be associated with lower CRP levels [80], a lower risk of IBD [81], and gut bacterial composition and diversity [82]. In contrast, in the interaction effect analysis, even when frequencies of fruit and vegetable intakes were added as independent variables, similar interaction effects between environmental sensitivity and the specific taxa in the gut microbiome were observed (see Appendix A for details). Finally, we could not explore the precise mechanism involved due to the nature of this cross-sectional study. However, based on our results, we speculated that short-chain fatty acids (such as butyric, propionic, and acetic acids) produced by the gut microbiome and/or the factors involved in the HPA axis, such as cortisol, might be involved in the interaction. Future studies could strengthen our findings by measuring levels of short-chain fatty acids (such as butyric, propionic, and acetic acids) in fecal samples and/or cortisol levels in samples, such as hair.

## 6. Conclusions

As a preliminary study, the present research suggests that the family *Marinifilaceae* (genus *Butyricimonas*), family *Barnesiellaceae* (genus *Coprobacter*), family *Akkermansiaceae* (genus *Akkermansia*), genus *Family XIII AD3011 group*, genus *GCA-900066225*, and genus *Ruminiclostridium 1* are key taxa in the gut microbiome associated with the relationship between environmental sensitivity and biomarkers of inflammation or gut permeability. Specifically, for individuals with high environmental sensitivity, these taxa might serve as one of the protective factors against inflammation and intestinal permeability. Further in-depth studies are required to confirm these associations and elucidate the underlying mechanisms.

## Figures and Tables

**Figure 1 microorganisms-13-00185-f001:**
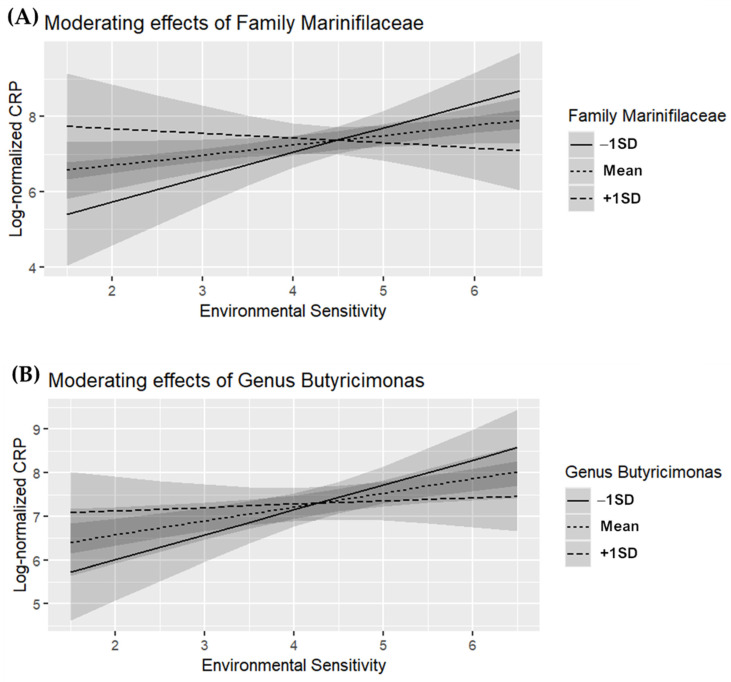
Moderating effects of gut microbiome taxa on the relationship between environmental sensitivity and C-reactive protein (CRP) (*n* = 88). (**A**) Family *Marinifilaceae*, (**B**) genus *Butyricimonas*.

**Figure 2 microorganisms-13-00185-f002:**
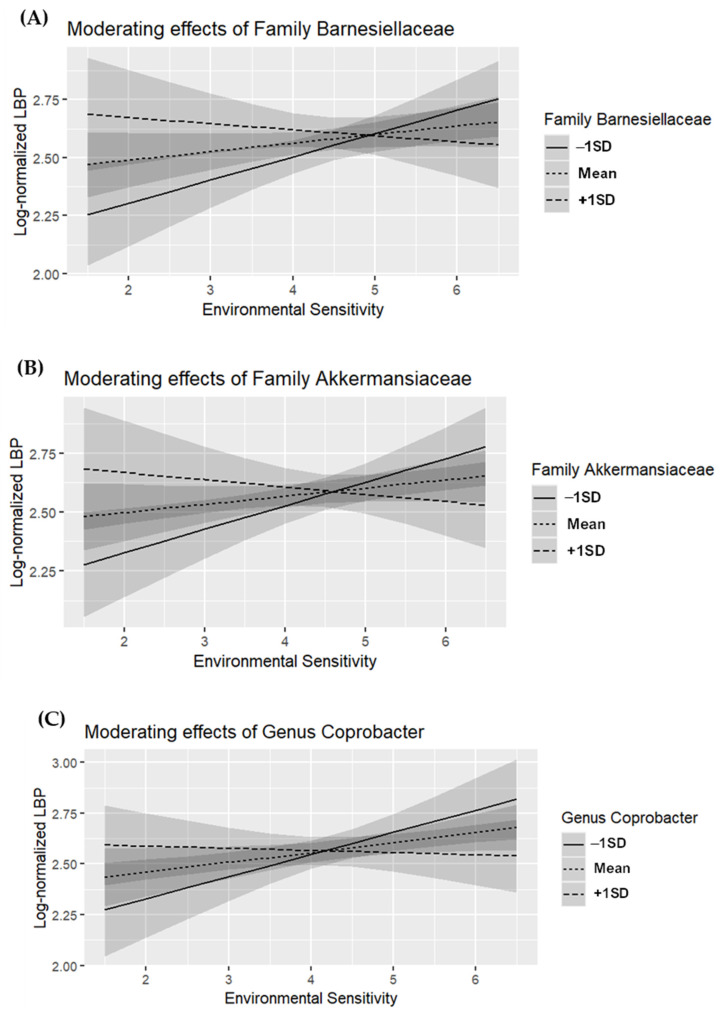
Moderating effects of gut microbiome taxa on the relationship between environmental sensitivity and lipopolysaccharide-binding protein (LBP) (*n* = 88). (**A**) Family *Barnesiellaceae*, (**B**) family *Akkermansiaceae*, (**C**) genus *Coprobacter*, (**D**) genus *Akkermansia*, (**E**) genus *Family XIII AD3011 group*, (**F**) genus *GCA-900066225*, (**G**) genus *Ruminiclostridium 1*.

**Table 1 microorganisms-13-00185-t001:** Clinical and biochemical characteristics of the subjects.

	Mean ± SD	95% CI
Age (years)	42 ± 10	40–44
Sex (female/male)	44/44 (50%/50%)	
Body weight (kg)	62.6 ± 12.7	60.0–65.3
Height (m)	1.65 ± 0.08	1.63–1.67
BMI (kg/m^2^)	22.8 ± 3.4	22.1–23.5
Environmental sensitivity	4.4 ± 1.0	4.2–4.6
CRP (ng/mL)	7994 ± 31040	1509–14480
LBP (µg/mL)	13.6 ± 4.0	12.8–14.5
Log-normalized CRP	7.33 ± 1.45	7.03–7.64
Log-normalized LBP	2.58 ± 0.26	2.52–2.63

Note. SD: standard deviation; CI: confidence interval; BMI: body mass index; CRP: C-reactive protein, LBP: lipopolysaccharide-binding protein.

**Table 2 microorganisms-13-00185-t002:** Relative abundance of the gut microbiome at the family level and Pearson correlation coefficients between alpha diversity indices and gut microbiome taxa.

	Prevalence (%)	Relative Abundance (%)			Correlation Coefficients			
					OTUs		PD	
		Mean		SD	*r*		*r*	
Family								
*Methanobacteriaceae*	8.0	0.035	±	0.182	0.380	**	0.399	**
*Actinomycetaceae*	93.2	0.050	±	0.037	−0.193	†	−0.138	
*Bifidobacteriaceae*	98.9	15.542	±	10.569	−0.197	†	−0.199	†
*Corynebacteriaceae*	15.9	0.003	±	0.015	−0.020		−0.039	
*Micrococcaceae*	28.4	0.005	±	0.014	−0.164		−0.159	
*Atopobiaceae*	39.8	0.086	±	0.220	0.105		0.081	
*Coriobacteriaceae*	89.8	6.444	±	4.769	0.004		−0.047	
*Coriobacteriales Incertae Sedis*	38.6	0.052	±	0.115	0.405	**	0.341	**
*Eggerthellaceae*	98.9	0.728	±	0.570	0.271	*	0.203	†
*Coriobacteriales unclassified family*	20.5	0.036	±	0.121	0.320	**	0.265	*
*Bacteroidaceae*	100.0	11.113	±	6.507	−0.217	*	−0.115	
*Barnesiellaceae*	64.8	0.420	±	0.603	0.323	**	0.361	**
*Marinifilaceae*	81.8	0.161	±	0.218	0.510	**	0.485	**
*Muribaculaceae*	18.2	0.085	±	0.297	0.120		0.075	
*Porphyromonadaceae*	8.0	0.002	±	0.009	0.134		0.136	
*Prevotellaceae*	60.2	1.561	±	3.280	−0.059		−0.143	
*Rikenellaceae*	87.5	0.923	±	0.890	0.613	**	0.586	**
*Tannerellaceae*	90.9	1.444	±	2.328	−0.122		−0.164	
*Bacteroidia unclassified order unclassified family*	6.8	0.004	±	0.020	0.277	**	0.288	**
*Campylobacteraceae*	5.7	0.004	±	0.025	0.000		0.083	
*Bacillaceae*	47.7	0.278	±	0.838	0.058		0.055	
*Bacillales Family XI*	53.4	0.011	±	0.019	−0.199	†	−0.160	
*Aerococcaceae*	5.7	0.001	±	0.002	−0.097		−0.127	
*Carnobacteriaceae*	48.9	0.010	±	0.013	−0.280	**	−0.256	*
*Enterococcaceae*	27.3	0.021	±	0.059	−0.261	*	−0.274	**
*Lactobacillaceae*	50.0	0.100	±	0.298	−0.148		−0.156	
*Leuconostocaceae*	9.1	0.009	±	0.048	0.281	**	0.210	*
*Streptococcaceae*	98.9	1.837	±	2.553	−0.248	*	−0.229	*
*Christensenellaceae*	65.9	0.532	±	1.069	0.669	**	0.609	**
*Clostridiaceae 1*	61.4	0.259	±	0.848	0.298	**	0.221	*
*Clostridiales vadin BB60 group*	26.1	0.013	±	0.038	0.519	**	0.509	**
*Defluviitaleaceae*	37.5	0.011	±	0.019	0.710	**	0.653	**
*Eubacteriaceae*	61.4	0.053	±	0.129	0.044		0.096	
*Clostridiales Family XI*	33.0	0.011	±	0.038	0.162		0.224	*
*Clostridiales Family XIII*	90.9	0.246	±	0.269	0.603	**	0.570	**
*Lachnospiraceae*	100.0	26.567	±	9.196	−0.359	**	−0.369	**
*Peptococcaceae*	36.4	0.029	±	0.120	0.276	**	0.258	*
*Peptostreptococcaceae*	97.7	1.497	±	1.864	0.283	**	0.225	*
*Ruminococcaceae*	100.0	18.208	±	8.074	0.691	**	0.639	**
*Clostridiales unclassified family*	13.6	0.003	±	0.012	0.434	**	0.405	**
*DTU014 uncultured bacterium*	11.4	0.002	±	0.007	0.477	**	0.491	**
*Erysipelotrichaceae*	100.0	2.792	±	2.801	0.185	†	0.200	†
*Acidaminococcaceae*	80.7	1.184	±	1.222	−0.196	†	−0.103	
*Veillonellaceae*	90.9	4.426	±	6.978	−0.286	**	−0.255	*
*Fusobacteriaceae*	43.2	0.168	±	0.479	−0.170		−0.117	
*Saccharimonadaceae*	22.7	0.003	±	0.007	−0.097		−0.106	
*Saccharimonadales uncultured bacterium*	5.7	0.001	±	0.003	0.151		0.158	
*Rhodospirillales uncultured bacterium*	14.8	0.028	±	0.151	0.182	†	0.178	†
*Desulfovibrionaceae*	81.8	0.129	±	0.168	0.236	*	0.274	**
*Succinivibrionaceae*	5.7	0.002	±	0.011	−0.101		−0.090	
*Burkholderiaceae*	86.4	0.232	±	0.362	−0.215	*	−0.077	
*Enterobacteriaceae*	81.8	0.349	±	0.691	−0.118		−0.017	
*Pasteurellaceae*	33.0	0.035	±	0.163	−0.092		−0.123	
*Synergistaceae*	18.2	0.020	±	0.119	0.370	**	0.351	**
*Izimaplasmatales unclassified family*	6.8	0.003	±	0.017	0.151		0.266	*
*Mollicutes RF39 uncultured bacterium*	9.1	0.006	±	0.028	0.425	**	0.400	**
*Akkermansiaceae*	56.8	2.211	±	4.247	0.279	**	0.333	**

Note. SD: standard deviation; OTUs: observed operational taxonomic units; PD: Faith’s phylogenetic diversity, ** *p* < 0.01; * *p* < 0.05; † *p* < 0.10.

**Table 3 microorganisms-13-00185-t003:** Relative abundance of the gut microbiome at the genus level and Pearson correlation coefficients between alpha diversity indices and gut microbiome taxa.

	Prevalence (%)	Relative Abundance (%)			Correlation Coefficients			
					OTUs		PD	
		Mean		SD	*r*		*r*	
Genus								
*Methanobrevibacter*	8.0	0.035	±	0.182	0.380	**	0.399	**
*Actinomyces*	93.2	0.048	±	0.036	−0.270	*	−0.201	†
*F0332*	5.7	0.001	±	0.002	0.018		0.009	
*Varibaculum*	6.8	0.002	±	0.007	0.302	**	0.235	*
*Bifidobacterium*	98.9	15.539	±	10.570	−0.197	†	−0.199	†
*Corynebacterium*	11.4	0.003	±	0.015	−0.047		−0.062	
*Rothia*	28.4	0.005	±	0.014	−0.164		−0.159	
*Atopobium*	11.4	0.001	±	0.004	−0.164		−0.154	
*Olsenella*	27.3	0.075	±	0.210	0.091		0.066	
*Collinsella*	89.8	6.410	±	4.777	−0.005		−0.054	
*Coriobacteriaceae unclassified genus*	8.0	0.033	±	0.206	0.204	†	0.152	
*Raoultibacter*	22.7	0.005	±	0.011	0.262	*	0.238	*
*Coriobacteriales Incertae Sedis uncultured bacterium*	28.4	0.047	±	0.112	0.386	**	0.324	**
*Adlercreutzia*	51.1	0.077	±	0.157	0.291	**	0.195	†
*Eggerthella*	88.6	0.337	±	0.375	−0.140		−0.099	
*Enterorhabdus*	23.9	0.036	±	0.090	0.312	**	0.291	**
*Gordonibacter*	56.8	0.034	±	0.058	0.271	*	0.288	**
*Senegalimassilia*	15.9	0.073	±	0.194	0.351	**	0.307	**
*Slackia*	28.4	0.108	±	0.337	0.020		−0.031	
*Eggerthellaceae uncultured bacterium*	12.5	0.049	±	0.158	0.234	*	0.162	
*Eggerthellaceae unclassified genus*	39.8	0.015	±	0.034	0.168		0.133	
*Coriobacteriales unclassified family unclassified genus*	20.5	0.036	±	0.121	0.320	**	0.265	*
*Bacteroides*	100.0	11.113	±	6.507	−0.217	*	−0.115	
*Barnesiella*	51.1	0.355	±	0.561	0.302	**	0.344	**
*Coprobacter*	43.2	0.058	±	0.141	0.138		0.142	
*Barnesiellaceae uncultured bacterium*	18.2	0.006	±	0.020	0.271	*	0.224	*
*Butyricimonas*	55.7	0.056	±	0.089	0.355	**	0.402	**
*Odoribacter*	78.4	0.106	±	0.168	0.473	**	0.417	**
*Muribaculaceae uncultured bacterium*	18.2	0.085	±	0.297	0.120		0.073	
*Porphyromonas*	8.0	0.002	±	0.009	0.134		0.136	
*Alloprevotella*	9.1	0.193	±	1.244	−0.160		−0.158	
*Paraprevotella*	20.5	0.203	±	0.566	−0.060		−0.078	
*Prevotella*	15.9	0.006	±	0.028	0.175		0.180	†
*Prevotella 2*	18.2	0.245	±	1.096	−0.114		−0.146	
*Prevotella 9*	23.9	0.808	±	2.228	0.054		−0.047	
*Alistipes*	86.4	0.914	±	0.895	0.613	**	0.586	**
*Parabacteroides*	90.9	1.444	±	2.328	−0.122		−0.164	
*Bacteroidia unclassified order unclassified family unclassified genus*	6.8	0.004	±	0.020	0.277	**	0.288	**
*Campylobacter*	5.7	0.004	±	0.025	0.000		0.083	
*Bacillus*	47.7	0.278	±	0.838	0.058		0.055	
*Gemella*	53.4	0.011	±	0.019	−0.199	†	−0.160	
*Abiotrophia*	5.7	0.001	±	0.002	−0.097		−0.127	
*Granulicatella*	48.9	0.010	±	0.013	−0.280	**	−0.256	*
*Enterococcus*	27.3	0.021	±	0.059	−0.261	*	−0.274	**
*Lactobacillus*	50.0	0.091	±	0.235	−0.134		−0.150	
*Leuconostoc*	8.0	0.006	±	0.038	0.274	**	0.212	*
*Lactococcus*	25.0	0.036	±	0.260	0.236	*	0.188	†
*Streptococcus*	98.9	1.801	±	2.555	−0.272	*	−0.247	*
*Christensenellaceae R-7 group*	59.1	0.492	±	1.026	0.657	**	0.597	**
*Christensenellaceae uncultured bacterium*	50.0	0.030	±	0.065	0.558	**	0.512	**
*Christensenellaceae unclassifie* *d genus*	44.3	0.010	±	0.016	0.394	**	0.376	**
*Clostridium sensu stricto 1*	61.4	0.259	±	0.848	0.298	**	0.221	*
*Clostridiales vadin BB60 group uncultured bacterium*	26.1	0.013	±	0.034	0.521	**	0.511	**
*Defluviitaleaceae UCG-011*	37.5	0.011	±	0.019	0.710	**	0.653	**
*Anaerofustis*	30.7	0.005	±	0.009	0.266	*	0.295	**
*Eubacterium*	37.5	0.048	±	0.127	0.027		0.079	
*Anaerococcus*	8.0	0.001	±	0.003	0.257	*	0.251	*
*Ezakiella*	10.2	0.002	±	0.009	0.132		0.145	
*Finegoldia*	5.7	0.001	±	0.002	−0.021		−0.038	
*Parvimonas*	15.9	0.005	±	0.025	0.113		0.184	†
*Peptoniphilus*	11.4	0.002	±	0.009	0.172		0.230	*
*Family XIII AD3011 group*	75.0	0.156	±	0.207	0.606	**	0.549	**
*Family XIII UCG-001*	56.8	0.029	±	0.039	0.464	**	0.459	**
*[Eubacterium] brachy group*	70.5	0.042	±	0.061	0.244	*	0.246	*
*[Eubacterium] nodatum group*	40.9	0.017	±	0.057	0.086		0.127	
*Anaerostipes*	100.0	1.914	±	1.880	−0.181	†	−0.198	†
*Blautia*	100.0	6.486	±	4.263	−0.253	*	−0.158	
*CAG-56*	21.6	0.047	±	0.133	0.196	†	0.157	
*Coprococcus 1*	35.2	0.104	±	0.185	0.327	**	0.242	*
*Coprococcus 2*	10.2	0.115	±	0.500	0.313	**	0.218	*
*Coprococcus 3*	42.0	0.346	±	0.600	0.086		0.012	
*Dorea*	79.5	1.331	±	1.363	−0.137		−0.163	
*Eisenbergiella*	50.0	0.052	±	0.119	0.212	*	0.323	**
*Fusicatenibacter*	85.2	2.052	±	2.045	−0.195	†	−0.290	**
*GCA900066575*	46.6	0.016	±	0.022	0.479	**	0.410	**
*Howardella*	5.7	0.006	±	0.031	0.147		0.159	
*Hungatella*	39.8	0.019	±	0.034	−0.165		−0.043	
*Lachnoclostridium*	100.0	1.356	±	1.096	−0.406	**	−0.286	**
*Lachnospira*	69.3	0.458	±	0.761	−0.230	*	−0.295	**
*Lachnospiraceae FCS020 group*	64.8	0.094	±	0.132	0.316	**	0.278	**
*Lachnospiraceae ND3007 group*	73.9	0.283	±	0.336	0.064		−0.036	
*Lachnospiraceae NK4A136 group*	77.3	0.391	±	0.737	0.302	**	0.271	*
*Lachnospiraceae UCG-001*	19.3	0.050	±	0.166	0.129		0.073	
*Lachnospiraceae UCG-004*	53.4	0.078	±	0.168	−0.179	†	−0.192	†
*Lachnospiraceae UCG-008*	14.8	0.004	±	0.012	0.148		0.077	
*Lachnospiraceae UCG-010*	36.4	0.008	±	0.016	−0.110		−0.143	
*Lactonifactor*	44.3	0.009	±	0.014	0.100		0.065	
*Marvinbryantia*	43.2	0.072	±	0.150	0.429	**	0.389	**
*Roseburia*	89.8	0.790	±	1.062	−0.106		−0.198	†
*Sellimonas*	65.9	0.266	±	0.432	−0.332	**	−0.267	*
*Shuttleworthia*	44.3	0.030	±	0.055	0.313	**	0.330	**
*Tyzzerella*	40.9	0.058	±	0.122	0.122		0.176	
*Tyzzerella 3*	27.3	0.107	±	0.274	−0.030		−0.073	
*Tyzzerella 4*	29.5	0.205	±	0.527	−0.262	*	−0.124	
*[Eubacterium] eligens group*	33.0	0.114	±	0.303	0.077		0.032	
*[Eubacterium] fissicatena group*	50.0	0.012	±	0.017	0.066		0.203	†
*[Eubacterium] hallii group*	88.6	1.587	±	1.355	0.020		−0.042	
*[Eubacterium] ruminantium group*	12.5	0.163	±	0.563	0.240	*	0.159	
*[Eubacterium] ventriosum group*	78.4	0.333	±	0.389	0.222	*	0.093	
*[Eubacterium] xylanophilum group*	14.8	0.014	±	0.056	0.162		0.218	*
*[Ruminococcus] gauvreauii group*	60.2	0.413	±	0.789	0.065		−0.002	
*[Ruminococcus] gnavus group*	86.4	1.248	±	2.124	−0.537	**	−0.387	**
*[Ruminococcus] torques group*	96.6	1.620	±	1.664	0.086		0.076	
*Lachnospiraceae uncultured bacterium*	96.6	0.622	±	0.592	−0.076		−0.100	
*Lachnospiraceae unclassified genus*	100.0	3.695	±	2.897	−0.094		−0.155	
*Peptococcus*	11.4	0.019	±	0.116	0.162		0.146	
*Peptococcaceae uncultured bacterium*	31.8	0.011	±	0.025	0.566	**	0.559	**
*Intestinibacter*	75.0	0.347	±	0.593	−0.026		−0.019	
*Peptostreptococcus*	9.1	0.001	±	0.004	−0.145		0.008	
*Romboutsia*	92.0	1.062	±	1.540	0.369	**	0.301	**
*Terrisporobacter*	18.2	0.036	±	0.149	0.097		0.055	
*Anaerofilum*	17.0	0.002	±	0.005	0.378	**	0.300	**
*Anaerotruncus*	51.1	0.015	±	0.021	0.413	**	0.461	**
*Butyricicoccus*	98.9	0.624	±	0.575	−0.346	**	−0.347	**
*Candidatus Soleaferrea*	12.5	0.003	±	0.010	−0.029		0.123	
*DTU089*	75.0	0.034	±	0.036	0.352	**	0.333	**
*Faecalibacterium*	96.6	4.966	±	3.755	−0.127		−0.148	
*Flavonifractor*	92.0	0.224	±	0.268	−0.247	*	−0.094	
*Fournierella*	20.5	0.011	±	0.044	−0.001		0.090	
*GCA-900066225*	53.4	0.017	±	0.031	0.426	**	0.422	**
*Hydrogenoanaerobacterium*	6.8	0.002	±	0.011	0.337	**	0.325	**
*Negativibacillus*	68.2	0.093	±	0.154	0.415	**	0.491	**
*Oscillibacter*	92.0	0.279	±	0.332	0.233	*	0.290	**
*Oscillospira*	10.2	0.011	±	0.061	0.093		0.061	
*Papillibacter*	8.0	0.001	±	0.004	0.314	**	0.290	**
*Ruminiclostridium*	47.7	0.011	±	0.021	0.452	**	0.379	**
*Ruminiclostridium 1*	9.1	0.001	±	0.005	0.368	**	0.299	**
*Ruminiclostridium 5*	98.9	0.852	±	1.256	0.284	**	0.217	*
*Ruminiclostridium 6*	13.6	0.173	±	1.006	0.127		0.180	†
*Ruminiclostridium 9*	83.0	0.213	±	0.215	0.156		0.251	*
*Ruminococcaceae NK4A214 group*	53.4	0.292	±	0.692	0.462	**	0.414	**
*Ruminococcaceae UCG-002*	59.1	0.704	±	1.170	0.550	**	0.544	**
*Ruminococcaceae UCG-003*	31.8	0.034	±	0.091	0.216	*	0.245	*
*Ruminococcaceae UCG-004*	25.0	0.130	±	0.239	0.124		0.207	†
*Ruminococcaceae UCG-005*	63.6	0.323	±	0.531	0.647	**	0.587	**
*Ruminococcaceae UCG-007*	8.0	0.001	±	0.004	0.426	**	0.416	**
*Ruminococcaceae UCG-009*	43.2	0.017	±	0.029	0.388	**	0.329	**
*Ruminococcaceae UCG-010*	31.8	0.040	±	0.109	0.606	**	0.533	**
*Ruminococcaceae UCG-013*	88.6	0.479	±	0.508	0.059		0.023	
*Ruminococcaceae UCG-014*	28.4	0.590	±	1.781	0.433	**	0.404	**
*Ruminococcus 1*	48.9	0.719	±	1.113	0.390	**	0.318	**
*Ruminococcus 2*	54.5	1.820	±	2.636	0.335	**	0.349	**
*Subdoligranulum*	95.5	3.194	±	2.771	0.150		0.084	
*UBA1819*	83.0	0.116	±	0.191	0.352	**	0.356	**
*[Eubacterium] coprostanoligenes group*	76.1	1.308	±	1.701	0.623	**	0.558	**
*Ruminococcaceae uncultured bacterium*	95.5	0.795	±	1.991	0.214	*	0.251	*
*Ruminococcaceae unclassified genus*	81.8	0.098	±	0.176	0.400	**	0.344	**
*Clostridiales unclassified family unclassified genus*	13.6	0.003	±	0.012	0.434	**	0.405	**
*DTU014 uncultured bacterium uncultured bacterium*	11.4	0.002	±	0.007	0.477	**	0.491	**
*Catenibacterium*	11.4	0.515	±	1.706	0.173		0.140	
*Catenisphaera*	12.5	0.059	±	0.315	0.193	†	0.144	
*Dielma*	13.6	0.002	±	0.007	0.106		0.107	
*Erysipelatoclostridium*	88.6	0.398	±	0.606	−0.127		−0.024	
*Erysipelotrichaceae UCG-003*	36.4	0.511	±	1.020	0.047		0.024	
*Faecalitalea*	35.2	0.109	±	0.297	−0.127		−0.070	
*Holdemanella*	21.6	0.820	±	2.032	0.054		0.079	
*Holdemania*	63.6	0.023	±	0.028	0.345	**	0.349	**
*Solobacterium*	23.9	0.014	±	0.105	−0.054		0.031	
*Turicibacter*	70.5	0.163	±	0.309	0.260	*	0.189	†
*[Clostridium] innocuum group*	81.8	0.055	±	0.072	−0.281	**	−0.139	
*Erysipelotrichaceae unclassified genus*	77.3	0.100	±	0.158	0.345	**	0.414	**
*Acidaminococcus*	35.2	0.309	±	0.820	−0.207	†	−0.221	*
*Phascolarctobacterium*	72.7	0.872	±	1.024	−0.068		0.054	
*Allisonella*	26.1	0.029	±	0.077	−0.152		−0.173	
*Dialister*	44.3	0.899	±	1.870	0.047		0.014	
*Megamonas*	26.1	1.667	±	6.081	−0.130		−0.093	
*Megasphaera*	35.2	0.946	±	2.058	−0.213	*	−0.199	†
*Mitsuokella*	12.5	0.321	±	1.466	−0.098		−0.104	
*Veillonella*	46.6	0.533	±	1.594	−0.413	**	−0.394	**
*Fusobacterium*	43.2	0.168	±	0.479	−0.170		−0.117	
*Saccharimonadaceae uncultured bacterium*	22.7	0.003	±	0.006	−0.185	†	−0.180	†
*Saccharimonadales uncultured bacterium uncultured bacterium*	5.7	0.001	±	0.003	0.151		0.158	
*Rhodospirillales uncultured bacterium uncultured bacterium*	14.8	0.028	±	0.151	0.182	†	0.178	†
*Bilophila*	78.4	0.083	±	0.117	0.024		0.074	
*Desulfovibrio*	26.1	0.040	±	0.105	0.302	**	0.295	**
*Desulfovibrionaceae uncultured bacterium*	13.6	0.005	±	0.019	0.261	*	0.307	**
*Parasutterella*	48.9	0.124	±	0.347	−0.171		−0.002	
*Sutterella*	67.0	0.104	±	0.178	−0.135		−0.181	†
*Burkholderiaceae unclassified genus*	5.7	0.003	±	0.015	0.313	**	0.282	**
*Citrobacter*	9.1	0.009	±	0.059	0.269	*	0.242	*
*Enterobacter*	10.2	0.031	±	0.163	−0.021		−0.078	
*Escherichia-Shigella*	71.6	0.267	±	0.584	−0.143		0.021	
*Raoultella*	5.7	0.006	±	0.048	−0.069		−0.094	
*Haemophilus*	33.0	0.035	±	0.163	−0.092		−0.123	
*Cloacibacillus*	17.0	0.019	±	0.119	0.369	**	0.350	**
*Izimaplasmatales unclassified family unclassified genus*	6.8	0.003	±	0.017	0.151		0.266	*
*Mollicutes RF39 uncultured bacterium uncultured bacterium*	9.1	0.006	±	0.028	0.425	**	0.400	**
*Akkermansia*	56.8	2.211	±	4.247	0.279	**	0.333	**

Note. SD: standard deviation; OTUs: observed operational taxonomic units; PD: Faith’s phylogenetic diversity, ** *p* < 0.01; * *p* < 0.05; † *p* < 0.10.

**Table 4 microorganisms-13-00185-t004:** Interaction effects between environmental sensitivity and family *Marinifilaceae* predicting CRP (*n* = 88).

	Log-Normalized CRP			
	Step1		Step2	
Predictors	*β*	*p*	*β*	*p*
Age	−0.151		−0.162	†
Sex	−0.090		−0.109	
BMI	0.542	**	0.537	**
HSP-J10	0.166	†	0.162	†
Family *Marinifilaceae*	0.015		0.032	
HSP-J10 × Family *Marinifilaceae*			−0.183	*
*R* ^2^	0.342	**	0.375	**
Δ*R*^2^			0.033	*

Note. ** *p* < 0.01; * *p* < 0.05; † *p* < 0.10. CRP: C-reactive protein; BMI: body mass index; HSP-J10: environmental sensitivity.

**Table 5 microorganisms-13-00185-t005:** Interaction effects between environmental sensitivity and family *Barnesiellaceae* predicting LBP (*n* = 88).

	Log-Normalized LBP			
	Step1		Step2	
Predictors	*β*	*p*	*β*	*p*
Age	0.041		0.059	
Sex	−0.035		−0.041	
BMI	0.507	**	0.443	**
HSP-J10	0.140		0.122	
Family *Barnesiellaceae*	0.126		0.136	
HSP-J10 × Family *Barnesiellaceae*			−0.200	*
*R* ^2^	0.304	**	0.340	**
Δ*R*^2^			0.036	*

Note. ** *p* < 0.01; * *p* < 0.05. LBP: lipopolysaccharide-binding protein; BMI: body mass index; HSP-J10: environmental sensitivity.

**Table 6 microorganisms-13-00185-t006:** Interaction effects between environmental sensitivity and family *Akkermansiaceae* predicting LBP (*n* = 88).

	Log-Normalized LBP			
	Step1		Step2	
Predictors	*β*	*p*	*β*	*p*
Age	0.042		0.033	
Sex	−0.021		−0.031	
BMI	0.506	**	0.526	**
HSP-J10	0.134		0.109	
Family *Akkermansiaceae*	0.028		0.058	
HSP-J10 × Family *Akkermansiaceae*			−0.192	*
*R* ^2^	0.290	**	0.325	**
Δ*R*^2^			0.035	*

Note. ** *p* < 0.01; * *p* < 0.05. LBP: lipopolysaccharide-binding protein; BMI: body mass index; HSP-J10: environmental sensitivity.

**Table 7 microorganisms-13-00185-t007:** Interaction effects between environmental sensitivity and family *Marinifilaceae* predicting LBP (*n* = 88).

	Log-Normalized LBP			
	Step1		Step2	
Predictors	*β*	*p*	*β*	*p*
Age	0.034		0.023	
Sex	−0.001		−0.019	
BMI	0.504	**	0.499	**
HSP-J10	0.129		0.126	
Family *Marinifilaceae*	−0.050		−0.034	
HSP-J10 × Family *Marinifilaceae*			−0.175	†
*R* ^2^	0.291	**	0.321	**
Δ*R*^2^			0.030	†

Note. ** *p* < 0.01; † *p* < 0.10. LBP: lipopolysaccharide-binding protein; BMI: body mass index; HSP-J10: environmental sensitivity.

**Table 8 microorganisms-13-00185-t008:** Interaction effects between environmental sensitivity and family *Defluviitaleaceae* predicting LBP (*n* = 88).

	Log-Normalized LBP			
	Step1		Step2	
Predictors	*β*	*p*	*β*	*p*
Age	0.041		0.038	
Sex	−0.035		−0.046	
BMI	0.533	**	0.538	**
HSP-J10	0.143		0.119	
Family *Defluviitaleaceae*	0.101		0.095	
HSP-J10 × Family *Defluviitaleaceae*			−0.166	†
*R* ^2^	0.298	**	0.324	**
Δ*R*^2^			0.027	†

Note. ** *p* < 0.01; † *p* < 0.10. LBP: lipopolysaccharide-binding protein; BMI: body mass index; HSP-J10: environmental sensitivity.

**Table 9 microorganisms-13-00185-t009:** Interaction effects between environmental sensitivity and family *Family XIII* predicting LBP (*n* = 88).

	Log-Normalized LBP			
	Step1		Step2	
Predictors	*β*	*p*	*β*	*p*
Age	0.043		0.015	
Sex	0.027		0.022	
BMI	0.501	**	0.516	**
HSP-J10	0.142		0.127	
Family *Family XIII*	−0.114		−0.043	
HSP-J10 × Family *Family XIII*			−0.169	†
*R* ^2^	0.300	**	0.323	**
Δ*R*^2^			0.023	†

Note. ** *p* < 0.01; † *p* < 0.10. LBP: lipopolysaccharide-binding protein; BMI: body mass index; HSP-J10: environmental sensitivity.

**Table 10 microorganisms-13-00185-t010:** Interaction effects between environmental sensitivity and genus *Butyricimonas* predicting CRP (*n* = 88).

	Log-Normalized CRP			
	Step1		Step2	
Predictors	*β*	*p*	*β*	*p*
Age	−0.152		−0.141	
Sex	−0.089		−0.106	
BMI	0.541	**	0.516	**
HSP-J10	0.165	†	0.206	*
Genus *Butyricimonas*	0.005		−0.035	
HSP-J10 × Genus *Butyricimonas*			−0.218	*
*R* ^2^	0.342	**	0.385	**
Δ*R*^2^			0.044	*

Note. ** *p* < 0.01; * *p* < 0.05; † *p* < 0.10. CRP: C-reactive protein; BMI: body mass index; HSP-J10: environmental sensitivity.

**Table 11 microorganisms-13-00185-t011:** Interaction effects between environmental sensitivity and genus *Coprobacter* predicting LBP (*n* = 88).

	Log-Normalized LBP			
	Step1		Step2	
Variables	*β*	*p*	*Β*	*p*
Age	0.041		−0.005	
Sex	−0.014		−0.018	
BMI	0.507	**	0.484	**
HSP-J10	0.134		0.163	†
Genus *Coprobacter*	0.019		−0.077	
HSP-J10 × Genus *Coprobacter*			−0.233	*
*R* ^2^	0.289	**	0.331	**
Δ*R*^2^			0.042	*

Note. ** *p* < 0.01; * *p* < 0.05; † *p* < 0.10. LBP: lipopolysaccharide-binding protein; BMI: body mass index; HSP-J10: environmental sensitivity.

**Table 12 microorganisms-13-00185-t012:** Interaction effects between environmental sensitivity and genus *Barnesiella* predicting LBP (*n* = 88).

	Log-Normalized LBP			
	Step1		Step2	
Predictors	*β*	*p*	*β*	*p*
Age	0.037		0.063	
Sex	−0.030		−0.030	
BMI	0.511	**	0.456	**
HSP-J10	0.138		0.112	
Genus *Barnesiella*	0.130		0.147	
HSP-J10 × Genus *Barnesiella*			−0.175	†
*R* ^2^	0.305	**	0.332	**
Δ*R*^2^			0.027	†

Note. ** *p* < 0.01; † *p* < 0.10. LBP: lipopolysaccharide-binding protein; BMI: body mass index; HSP-J10: environmental sensitivity.

**Table 13 microorganisms-13-00185-t013:** Interaction effects between environmental sensitivity and genus *Akkermansia* predicting LBP (*n* = 88).

	Log-Normalized LBP			
	Step1		Step2	
Predictors	*β*	*p*	*β*	*p*
Age	0.042		0.033	
Sex	−0.021		−0.031	
BMI	0.506	**	0.526	**
HSP-J10	0.134		0.109	
Genus *Akkermansia*	0.028		0.058	
HSP-J10 × Genus *Akkermansia*			−0.192	*
*R* ^2^	0.290	**	0.325	**
Δ*R*^2^			0.035	*

Note. ** *p* < 0.01; * *p* < 0.05. LBP: lipopolysaccharide-binding protein; BMI: body mass index; HSP-J10: environmental sensitivity.

**Table 14 microorganisms-13-00185-t014:** Interaction effects between environmental sensitivity and genus *Family XIII AD3011 group* predicting LBP (*n* = 88).

	Log-Normalized LBP			
	Step1		Step2	
Predictors	*β*	*p*	*β*	*p*
Age	0.045		−0.014	
Sex	0.015		0.020	
BMI	0.493	**	0.556	**
HSP-J10	0.148		0.134	
Genus *Family XIII AD3011 group*	−0.102		0.065	
HSP-J10 × Genus *Family XIII AD3011 group*			−0.284	*
*R* ^2^	0.298	**	0.351	**
Δ*R*^2^			0.053	*

Note. ** *p* < 0.01; * *p* < 0.05. LBP: lipopolysaccharide-binding protein; BMI: body mass index; HSP-J10: environmental sensitivity.

**Table 15 microorganisms-13-00185-t015:** Interaction effects between environmental sensitivity and genus *GCA-900066225* predicting LBP (*n* = 88).

	Log-Normalized LBP			
	Step1		Step2	
Predictors	*β*	*p*	*β*	*p*
Age	0.031		0.001	
Sex	0.004		−0.025	
BMI	0.500	**	0.521	**
HSP-J10	0.148		0.091	
Genus *GCA-900066225*	−0.121		−0.050	
HSP-J10 × Genus *GCA-900066225*			−0.228	*
*R* ^2^	0.303	**	0.346	**
Δ*R*^2^			0.043	*

Note. ** *p* < 0.01; * *p* < 0.05. LBP: lipopolysaccharide-binding protein; BMI: body mass index; HSP-J10: environmental sensitivity.

**Table 16 microorganisms-13-00185-t016:** Interaction effects between environmental sensitivity and genus *Ruminiclostridium 1* predicting LBP (*n* = 88).

	Log-Normalized LBP			
	Step1		Step2	
Predictors	*β*	*p*	*β*	*p*
Age	0.039		0.047	
Sex	−0.012		−0.037	
BMI	0.511	**	0.508	**
HSP-J10	0.134		0.115	
Genus *Ruminiclostridium 1*	0.016		−0.068	
HSP-J10 × Genus *Ruminiclostridium 1*			−0.208	*
*R* ^2^	0.289	**	0.324	**
Δ*R*^2^			0.035	*

Note. ** *p* < 0.01; * *p* < 0.05. LBP: lipopolysaccharide-binding protein; BMI: body mass index; HSP-J10: environmental sensitivity.

## Data Availability

The original contributions presented in this study are included in the article/Appendix A. Further inquiries can be directed to the corresponding author.

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
