# Peer review of "Key Taxa of the Gut Microbiome Associated with the Relationship Between Environmental Sensitivity and Inflammation-Related Biomarkers"

_microorganisms, 2025, doi:10.3390/microorganisms13010185_

Round 1
Reviewer 1 Report (Previous Reviewer 2)
Comments and Suggestions for Authors
My previous comments were:
--------------------------General comments
The aim of the present study was to investigate the key taxa of the gut microbiome that influence the relationship between environmental sensitivity and inflammation biomarkers. The study is well written and carefully designed. The rationale is clear, and the statistical procedures adopted are consistent with the research question. However, minor adjustments are necessary.
--------------------------Specific comments
----------Introduction
-Line 41-42: "Several studies have found positive associations" but only one study was cited. Consider adding more references to support this statement.
----------Materials and Methods
-Line 76: Unnecessary. Remove this line as it does not add value to the section.
-Line 76-78: Something is unclear here. If the previous study involved 110 adults, but informed consent was obtained from 90 in this study, does it mean the previous study was published without the consent of 20 participants? Clarify this point.
-Avoid using subtopics for statistical analysis as it creates confusion and lacks coherence. For instance, beginning line 146 with "Second" makes little sense in this context.
----------Results
-Age should not have decimal places. Present it as whole numbers.
-Replace "gender" with "sex".
-Include mass and height data in Table 1 for completeness.
-Calculate and present confidence intervals for the data shown in Table 1.
-Shift the panel labels (letters) on the figures to the left for better alignment and clarity.
__________________________________________________________________________
The authors have revised the manuscript based on my recommendations. I have no further suggestions.
Author Response
Comment:
The authors have revised the manuscript based on my recommendations. I have no further suggestions.
Response:
Thank you very much for your thoughtful review and for taking the time to assess the revised manuscript. We truly appreciate your positive feedback and am glad to hear that the revisions align with your recommendations. Your input has been invaluable in enhancing the quality of the manuscript. Thank you once again for your support.
Reviewer 2 Report (Previous Reviewer 1)
Comments and Suggestions for Authors
The manuscript explores a topic of growing scientific interest, namely the role of the gut microbiota as a possible modulator of the effects of environmental sensitivity on inflammatory and intestinal permeability biomarkers. However, after a careful reading of this new version of the paper, some methodological and conceptual criticalities emerge that reduce the scientific impact and clinical applicability of the results. The manuscript offers interesting theoretical insights, but the study design and interpretation of the results do not fully justify the conclusions. In order to increase the impact of the work, a longitudinal or experimental approach would need to be taken and the biological mechanisms underlying the observations would need to be investigated in more detail. Currently, the manuscript does not meet the requirements for publication in a high-impact journal.
1. Limited originality of the results:
The conclusions of the manuscript are based on interventions that are already known to improve obesity, inflammation and gut health, such as increasing fibre consumption or weight control. This makes the role of the microbiota more an indicator of metabolic and inflammatory changes rather than an independent causal factor. Consequently, the added value of the microbiota as an analytical variable is reduced.
2. Limitations of the study design:
The study uses a cross-sectional approach that prevents causal relationships between environmental sensitivity, microbiota and biomarkers from being established. The associations described, although statistically significant, remain speculative. A longitudinal or interventional approach would have been useful to confirm the hypothesis that the identified taxa have a direct modulating effect.
3. Confounding factors:
Although the authors considered some confounding factors, such as BMI and frequency of fruit and vegetable consumption, it is unclear how other important elements, e.g. level of physical activity or chronic stress, were controlled for. These factors could partially explain the observed associations, reducing the impact attributable to the microbiota.
4. Insufficiently supported clinical implications:
The proposed therapeutic implications (use of prebiotics and probiotics) do not appear innovative or specifically justified by the study results. The link between microbiota taxa and biomarkers would require validation through direct interventions on the microbiota, which were not carried out.
5. Lack of depth in biological mechanisms:
Although hypotheses are proposed about the mechanisms underlying the observations (e.g. HPA axis, short-chain fatty acids), these remain speculative and lacking supporting experimental data.
Author Response
We would like to sincerely thank you for taking the time to review our manuscript again. Your valuable feedback and insights have been instrumental in enhancing the quality of this work. We appreciate your careful consideration and comments.
In light of your observations, we would like to provide some responses and clarifications regarding specific points raised in your review. We believe this will help to further clarify the intentions and findings of the study. Thank you once again for your support and guidance throughout this process.
Comments 1:
- Limited originality of the results:
The conclusions of the manuscript are based on interventions that are already known to improve obesity, inflammation and gut health, such as increasing fibre consumption or weight control. This makes the role of the microbiota more an indicator of metabolic and inflammatory changes rather than an independent causal factor. Consequently, the added value of the microbiota as an analytical variable is reduced.
Response 1:
Thank you for your comments regarding the originality of the results. It is important to note that our findings demonstrate that the interaction between environmental sensitivity and gut microbiome taxa serves as an independent predictor of CRP and LBP, even when adjusting for independent variables such as BMI and the frequency of vegetable and fruit intakes. Additionally, cross-sectional study designs are commonly used in this field of research, and we acknowledge their limitations in establishing causal relationships, which we have addressed in the “Possible limitations and future directions” section of the manuscript (Page 20, Line 466-470). Therefore, we believe that the presentation of our results and the discussions based on them are appropriate and provide valuable insights into this field of research.
Comments 2:
- Limitations of the study design:
The study uses a cross-sectional approach that prevents causal relationships between environmental sensitivity, microbiota and biomarkers from being established. The associations described, although statistically significant, remain speculative. A longitudinal or interventional approach would have been useful to confirm the hypothesis that the identified taxa have a direct modulating effect.
Response 2:
Thank you for your insightful feedback regarding the study design. As mentioned in the manuscript (Page 20, Line 467-468), cross-sectional study is a commonly used research design in this field, and many studies published in this journal also employ this study design (1-5). Our conclusions are carefully limited to the scope of findings that can be drawn from cross-sectional study design. We have also emphasized the need for a longitudinal approach in future research directions (Page 20, Line 469-470). Therefore, we believe that the design and conclusions of this study are appropriate and contribute valuable insights to this field of research.
(1) Vu et al., Microorganisms 2021, 9, doi:10.3390/microorganisms9102115.
(2) Kashtanova, et al., Microorganisms 2020, 8, 1162.
(3) Park et al., Microorganisms. 2023, 11, 1892. doi: 10.3390/microorganisms11081892.
(4) Shields et al., Microorganisms. 2023, 11, 1405. doi: 10.3390/microorganisms11061405.
(5) Kashtanova et al., Microorganisms. 2020, 8, 1162. doi: 10.3390/microorganisms8081162.
Comments 3:
- Confounding factors:
Although the authors considered some confounding factors, such as BMI and frequency of fruit and vegetable consumption, it is unclear how other important elements, e.g. level of physical activity or chronic stress, were controlled for. These factors could partially explain the observed associations, reducing the impact attributable to the microbiota.
Response 3:
Thank you for your comment regarding confounding factors. In general, it is essential to select independent variables appropriately depending on the sample size, as adding too many independent variables can be statistically irrelevance. In this study, we have chosen to include only important variables. Furthermore, we have also conducted additional analyses that included dietary factors, such as vegetable and fruit intakes, as independent variables. Therefore, we believe that our selection of independent variables is appropriate for this research.
Comments 4:
- Insufficiently supported clinical implications:
The proposed therapeutic implications (use of prebiotics and probiotics) do not appear innovative or specifically justified by the study results. The link between microbiota taxa and biomarkers would require validation through direct interventions on the microbiota, which were not carried out.
Response 4:
Thank you for your constructive feedback regarding the clinical implications. Indeed, what we presented in the clinical implications section may be an overstatement given the results of this study. Therefore, we have decided to remove the clinical implications from the manuscript (Page 20). Thank you for pointing this out.
Comments 5:
- Lack of depth in biological mechanisms:
Although hypotheses are proposed about the mechanisms underlying the observations (e.g. HPA axis, short-chain fatty acids), these remain speculative and lacking supporting experimental data.
Response 5:
Thank you for your thoughtful comments regarding the discussion of biological mechanisms. This study employs a cross-sectional design, which is commonly used in this field, and it is explicitly noted in the manuscript that this study design does not allow for the evaluation of mechanisms (Page 21, Line 477-478). Investigating mechanisms was not the primary objective of this study; however, we believe it is essential to explore potential mechanisms for our findings. Therefore, we have included a discussion of these possibilities along with insights into future directions for research. However, the discussion regarding SCFAs may be overstated. We have revised the relevant sections to present a more concise and balanced perspective (Page 20, Line 436-438). We believe that this approach is sufficient for the scope of this manuscript. Thank you for your understanding.

Reviewer 3 Report (New Reviewer)
Comments and Suggestions for Authors
In this study, Takasugi et al. further investigated the key gut microbiome taxa associated with the relationship between environmental sensitivity and inflammation-related biomarkers in 88 participants based on their previous study. The results showed that specific taxa, such as Butyricimonas, Coprobacter, and Akkermansia, played a protective role against inflammation and gut permeability in individuals with high environmental sensitivity. Despite some of the analysis conducted in this study, several points need further clarification. Here are some comments on this study:
1. In the gut microbiome section of the method, it is recommended that the authors provide detailed methods and a taxonomy database for 16S rRNA gene data processing.
2. Line 148 “Following this, interaction effect analysis was performed at the genus level only 148 for those bacterial taxa that exhibited significant interactions at the family level”, I personally believe it would be more reasonable to do the analysis of all genera.
3. There were 44 males and 44 females. Is there a gender difference in gut microbiome?
4. Table 2 Relative abundance (%) results appear to be incorrect, please check it.
5. It would be useful for the authors to explain the reasons why the abundance of families or genera should be related to the diversity of the gut microbiome. What does this relationship represent? The abundance of families or genera and the diversity are both obtained from the OTUs abundance results. Personally, I don't think the two results (abundance and diversity) are independent.
6. In the manuscript and in supplementary files, there are a lot of yellow highlighted marks, and it is not clear what they mean.
7. I consider the speculation and discussion about short-chain fatty acids (SCFAs) to be a bit far-fetched and overstated without the experimental results.
Author Response
Thank you for your constructive feedback and suggestions. We appreciate the time and effort you have dedicated to reviewing our paper. In light of your comments, we have revised the manuscript.
Comments 1:
- In the gut microbiome section of the method, it is recommended that the authors provide detailed methods and a taxonomy database for 16S rRNA gene data processing.
Response 1:
We appreciate these helpful suggestions. As you suggested, we have added the detailed methods and a taxonomy database for 16S rRNA gene data processing in the “Materials and Methods” section (Page 3, Line 128-136).
Comments 2:
- Line 148 “Following this, interaction effect analysis was performed at the genus level only 148 for those bacterial taxa that exhibited significant interactions at the family level”, I personally believe it would be more reasonable to do the analysis of all genera.
Response 2:
We appreciate your suggestions. In fact, since some families contain various kinds of genus, it was thought that there might be a bias when interaction effect analysis was performed at the genus level only for those bacterial taxa that exhibited significant interactions at the family level. Therefore, in addition to the genus that exhibited significant interaction at the family level, we also analyzed the interactions of taxa genus that showed significant moderate (|r| > 0.30) or high (|r| > 0.50) correlations with alpha diversity indices of the gut microbiota (Page 4, Line 149, 155-157; Page 5, Line 205-224; Table 3; Page 13; Line 285-288, 301-304; Page 15, Line 343-344). This analysis showed that for CRP, additional significant predictor was not observed and for LBP, similar findings were observed for genus Family XIII AD3011 group, genus GCA-900066225, and genus Ruminiclostridium 1. Therefore, we added the results and discussion about these taxa in “Abstract” section (Page 1, Line 20-21), “Results” section (Page 13, Line 311-313; Tables 14-16; Page 16, Line 364-374; Figures 2E, 2F, 2G), “Discussion” section (Page 19, Line397-398; Page 20, Line 449-457), “Conclusions” section (Page 21, Line 486-487) and “Supplementary materials” (Supplementary Tables 8-10).
Comments 3:
- There were 44 males and 44 females. Is there a gender difference in gut microbiome?
Response 3:
We appreciate your insightful advice. We have tested the gender differences in gut microbiome taxa (Page 3, Line 143-144) and have added the results (Page 4, 5, Line183-198).
Comments 4:
- Table 2 Relative abundance (%) results appear to be incorrect, please check it.
Response 4:
Thank you for your comment. We have thoroughly checked the relative abundance results presented in Table 2 and confirmed that the results were correct. In Tables 2 (and 3), the results exclude taxa with a prevalence of less than 5%, which is why the total sum of relative abundance does not equal 100%.
Comments 5:
- It would be useful for the authors to explain the reasons why the abundance of families or genera should be related to the diversity of the gut microbiome. What does this relationship represent? The abundance of families or genera and the diversity are both obtained from the OTUs abundance results. Personally, I don't think the two results (abundance and diversity) are independent.
Response 5:
Thank you for your insightful comment. In our previous study, we found that the interactions between environmental sensitivity and gut microbiome diversity indices accounted for the levels of CRP and LBP, suggesting that gut microbiome diversity may serve a protective function against inflammatory responses in individuals with high environmental sensitivity. Based on these findings, this study focused on taxa correlated with gut microbiome diversity in order to identify specific taxa among them. Therefore, as you correctly pointed out, the gut bacterial taxa identified in this study are closely associated with gut microbiome diversity.
Comments 6:
- In the manuscript and in supplementary files, there are a lot of yellow highlighted marks, and it is not clear what they mean.
Response 6:
The manuscript was a resubmitted version to this journal. The highlighted yellow marks in the manuscript and supplementary file indicated the changes and additions made since the previous submission.
Comments 7:
- I consider the speculation and discussion about short-chain fatty acids (SCFAs) to be a bit far-fetched and overstated without the experimental results.
Response 7:
Thank you for your valuable feedback. We agree that the discussion regarding SCFAs was overstated. We have revised the relevant sections to present a more concise and balanced perspective (Page 20, Line 436-438).

Round 2
Reviewer 2 Report (Previous Reviewer 1)
Comments and Suggestions for Authors
The manuscript presents a study on the gut microbiota, exploring associations between environmental sensitivity, bacterial taxa and inflammatory biomarkers . Despite the potential importance of the topic, the work does not make significant contributions to the field. The results merely describe associations already well documented in the literature and provide no new mechanistic insights or practical applications. In addition, the cross-sectional design of the study severely limits the ability to establish causality, leaving conclusions speculative.
Main Comments.
The originality of the results is limited, with associations already well documented and little innovation from previous studies. The microbiota emerges as a marginal variable in the context of inflammation and environmental sensitivity, making no new contributions. The cross-sectional design does not allow causality to be established, and the authors do not present concrete strategies to overcome this limitation, making the conclusions purely speculative. Furthermore, although some confounding factors such as BMI and fruit and vegetable consumption were controlled for, crucial variables such as physical activity and chronic stress, which could influence the observed associations, are missing. Clinical implications related to the use of prebiotics and probiotics are weak and not supported by the results, highlighting the lack of practical applicability of the study. Finally, hypotheses about biological mechanisms remain speculative and unsupported by experimental data, reducing the impact of the work. In conclusion, the manuscript does not make a significant contribution to the understanding of the role of the gut microbiota in modulating environmental sensitivity and inflammatory biomarkers. Methodological limitations and lack of originality make the work unsuitable for publication in this form.
I suggest the authors consider a longitudinal or interventional study to validate their hypotheses and integrate functional or experimental data to explore the underlying mechanisms. Only with a more robust and innovative approach could the work make a significant contribution to the field.
Author Response
Response:
Thank you for your thorough review of our manuscript and for providing valuable feedback. We appreciate your insights and would like to address your concerns.
We would like to emphasize once again that cross-sectional studies are commonly utilized in this research field to establish preliminary associations and generate hypotheses for further investigation. We acknowledge the limitations inherent in our study design, and we have already added explicit mentions of these limitations in our manuscript. Although this study has several limitations, we believe that our findings are valuable and deserve to be shared with other scientists working on similar issues. In light of these points, we have revised the Abstract, Introduction, and Conclusion sections (Page 1, Line 14, 25-26; Page 2, Line 75; Page 21, Line 486, 492-494) to clarify that these data should be considered as preliminary and possibly preparatory for more in-depth studies to be conducted in the future.
Thank you again for your valuable feedback, which has helped us refine our manuscript.

Reviewer 3 Report (New Reviewer)
Comments and Suggestions for Authors
I thank the authors for addressing all my comments
Author Response
Response:
Thank you very much for your thoughtful review and for taking the time to assess the revised manuscript. We truly appreciate your positive feedback and are glad to hear that the revisions align with your comments. Your input has been invaluable in enhancing the quality of the manuscript. Thank you once again for your support.

This manuscript is a resubmission of an earlier submission. The following is a list of the peer review reports and author responses from that submission.
Round 1
Reviewer 1 Report
Comments and Suggestions for Authors
The paper addresses an interesting topic by examining the interaction between environmental sensitivity and the gut microbiota in relation to inflammatory markers such as CRP and LBP. However, there are some critical points that need to be considered to improve the quality and interpretation of the results.
1. Absence of dietary controls and confounding factors
The composition of the gut microbiota is strongly influenced by diet, which was not taken into account in this study. Diet has a significant impact on the main taxa of interest, which could alter the observed relationship with inflammatory biomarkers. Including dietary data or at least discussing this as a potential limitation is essential for an accurate assessment of the correlations between environmental sensitivity and microbiota.
2. Cross-sectional design
The study design, being cross-sectional, limits the ability to determine causality between environmental sensitivity, microbiota and inflammation. It would be useful to suggest future longitudinal studies to better explore these causal relationships.
3. Exclusion of unclassified taxa
The exclusion of unclassified or uncultured taxa may have omitted important microbial groups, potentially affecting the results. This point should be explored further in the limitations section.
4. Physiological mechanisms
Although reference is made to potential mechanisms (such as butyrate production and gut barrier integrity), the paper does not provide data on short-chain fatty acid levels, which could help better support the hypotheses. Incorporating these measures or proposing them in future studies would greatly strengthen the conclusions.
5. Validity of environmental sensitivity assessment instruments
The exclusive use of questionnaires to assess environmental sensitivity, without considering other objective indicators (such as neurophysiological tests or genetic data), is a limitation. The integration of alternative methods could offer a more comprehensive view.
6. Conclusions not fully supported
Conclusions concerning the protective role of certain microbiotic taxa appear forced without more attention to other determinants, such as diet and the use of antibiotics or probiotics. I recommend reformulating the conclusions, recognising that the role of the microbiota is only one part of a more complex picture.
Author Response
Thank you for your constructive feedback and suggestions. We appreciate the time and effort you have put into reviewing our paper. According to your comments, we revised the manuscript.
- Absence of dietary controls and confounding factors
The composition of the gut microbiota is strongly influenced by diet, which was not taken into account in this study. Diet has a significant impact on the main taxa of interest, which could alter the observed relationship with inflammatory biomarkers. Including dietary data or at least discussing this as a potential limitation is essential for an accurate assessment of the correlations between environmental sensitivity and microbiota.
Reply: As you pointed out, our date cannot exclude the possibility that there might be other confounding factors such as diet influencing the results. We included this point in the Limitations section (Line 413-415, in the revised manuscript).
- Cross-sectional design
The study design, being cross-sectional, limits the ability to determine causality between environmental sensitivity, microbiota and inflammation. It would be useful to suggest future longitudinal studies to better explore these causal relationships.
Reply: As you suggested, we added the sentence, “Future longitudinal studies would allow for better exploration of the causal relationships” in the Limitations section (Line 409-410 in the revised manuscript).
- Exclusion of unclassified taxa
The exclusion of unclassified or uncultured taxa may have omitted important microbial groups, potentially affecting the results. This point should be explored further in the limitations section.
Reply: As you suggested, we edited the text (Line 411 in the revised manuscript) and added the sentence, “In the future, when databases on uncultured or unclassified taxa are constructed, it would be worthwhile to conduct additional analyses.” in the Limitations section (Line 411-413 in the revised manuscript).
- Physiological mechanisms
Although reference is made to potential mechanisms (such as butyrate production and gut barrier integrity), the paper does not provide data on short-chain fatty acid levels, which could help better support the hypotheses. Incorporating these measures or proposing them in future studies would greatly strengthen the conclusions.
Reply: As you pointed out, we added the sentence, “In the future studies, measuring levels of short-chain fatty acid (such as butyric, propionic, and acetic acids) in fecal samples will strengthen our findings.” in the Limitations section (Line 415-417 in the revised manuscript).
- Validity of environmental sensitivity assessment instruments
The exclusive use of questionnaires to assess environmental sensitivity, without considering other objective indicators (such as neurophysiological tests or genetic data), is a limitation. The integration of alternative methods could offer a more comprehensive view.
Reply: According to your suggestion, we edited the text in the Limitations section (Line 405-408 in the revised manuscript).
- Conclusions not fully supported
Conclusions concerning the protective role of certain microbiotic taxa appear forced without more attention to other determinants, such as diet and the use of antibiotics or probiotics. I recommend reformulating the conclusions, recognising that the role of the microbiota is only one part of a more complex picture.
Reply: As you suggested, we referred the other possible confounding factors in the Limitations section (Line 413-415 in the revised manuscript) and added the sentence “Further studies are needed to elucidate the causal relationships.” in the conclusion section (Line 424-425 in the revised manuscript).
We hope these revisions will address your concerns and improve the quality of our manuscript. Once again, thank you for your valuable input.

Reviewer 2 Report
Comments and Suggestions for Authors
--------------------------General comments
The aim of the present study was to investigate the key taxa of the gut microbiome that influence the relationship between environmental sensitivity and inflammation biomarkers. The study is well written and carefully designed. The rationale is clear, and the statistical procedures adopted are consistent with the research question. However, minor adjustments are necessary.
--------------------------Specific comments
----------Introduction
-Line 41-42: "Several studies have found positive associations" but only one study was cited. Consider adding more references to support this statement.
----------Materials and Methods
-Line 76: Unnecessary. Remove this line as it does not add value to the section.
-Line 76-78: Something is unclear here. If the previous study involved 110 adults, but informed consent was obtained from 90 in this study, does it mean the previous study was published without the consent of 20 participants? Clarify this point.
-Avoid using subtopics for statistical analysis as it creates confusion and lacks coherence. For instance, beginning line 146 with "Second" makes little sense in this context.
----------Results
-Age should not have decimal places. Present it as whole numbers.
-Replace "gender" with "sex".
-Include mass and height data in Table 1 for completeness.
-Calculate and present confidence intervals for the data shown in Table 1.
-Shift the panel labels (letters) on the figures to the left for better alignment and clarity.
Author Response
Thank you for your constructive feedback and suggestions. We appreciate the time and effort you have put into reviewing our paper. According to your comments, we revised the manuscript.
----------Introduction
-Line 41-42: "Several studies have found positive associations" but only one study was cited. Consider adding more references to support this statement.
Reply: According to your suggestion, we added the following references:
Baliatsas, C.; van Kamp, I.; Hooiveld, M.; Lebret, E.; Yzermans, J. The relationship of modern health worries to non-specific physical symptoms and perceived environmental sensitivity: A study combining self-reported and general practice data. J Psychosom Res. 2015, 79, 355-361, doi:https://doi.org/10.1016/j.jpsychores.2015.09.004.
Le, T.L.; Geist, R.; Hunter, J.; Maunder, R.G. Relationship between insecure attachment and physical symptom severity is mediated by sensory sensitivity. Brain Behav 2020, 10, e01717, doi:10.1002/brb3.1717.
----------Materials and Methods
-Line 76: Unnecessary. Remove this line as it does not add value to the section.
Reply: According to your suggestion, we have modified this line (Line 78-80 in the revised manuscript).
-Line 76-78: Something is unclear here. If the previous study involved 110 adults, but informed consent was obtained from 90 in this study, does it mean the previous study was published without the consent of 20 participants? Clarify this point.
Reply: One hundred and ten participants had consented to participate in another study measuring fecal and/or blood biomarkers, and 90 of these participants consented to participate in this study and complete an additional questionnaire. We have updated the text to clarify this point (Line 78-81 in the revised manuscript).
-Avoid using subtopics for statistical analysis as it creates confusion and lacks coherence. For instance, beginning line 146 with "Second" makes little sense in this context.
Reply: According to your suggestion, we removed subtopics for statistical analysis.
----------Results
-Age should not have decimal places. Present it as whole numbers.
Reply: According to your suggestion, we presented age as whole number (Table 1).
-Replace "gender" with "sex".
Reply: According to your suggestion, we replace "gender" with "sex" (Line 162 in the revised manuscript, Table 1, Table 3 – 12, Supplementary Table 1).
-Include mass and height data in Table 1 for completeness.
Reply: According to your suggestion, we added body weight and height data in Table 1.
-Calculate and present confidence intervals for the data shown in Table 1.
Reply: According to your suggestion, we presented 95% confidence intervals for the data shown in Table 1.
-Shift the panel labels (letters) on the figures to the left for better alignment and clarity.
Reply: According to your suggestion, we shifted the panel labels on the figures to the left (Figure 1, 2).
We hope these revisions will address your concerns and improve the quality of our manuscript. Once again, thank you for your valuable input.

Round 2
Reviewer 1 Report
Comments and Suggestions for Authors
The authors have made only minor corrections to the paper.
1. Superficial response to major concerns: Although the authors acknowledge the limitations of their methods - dietary control, exclusion of unclassified taxa and reliance on cross-sectional data - they merely note these without taking substantial steps to address these shortcomings or re-evaluate the results based on alternative hypotheses.
2. Lack of depth in causal interpretations: The response to comments on causality in the study design is insufficient. Merely observing that ‘longitudinal studies are needed’ lacks the rigour expected in addressing this limitation for a study that draws strong conclusions about interactions between microbiota and environmental sensitivity.
3. Insufficient support for key mechanistic claims: The authors were advised to include additional physiological parameters, such as levels of short-chain fatty acids, that would strengthen the mechanism claims. Adding this limitation does not provide the necessary support for their mechanistic interpretations.